# THE SMALL BATCH SIZE ANOMALY IN MULTISTEP DEEP REINFORCEMENT LEARNING

## 1. Introduction

Deep reinforcement learning (DRL), which combines traditional reinforcement learning (RL) techniques with neural networks, has had a number of recent successes (Schrittwieser et al., 2020; Bellemare et al., 2020; Degrave et al., 2022). Yet successful application of DRL to new problems remains a challenge, in large part due to the difficulty in understanding how neural network training is affected by the vast number of hyper-parameters involved. Despite a number of recent works developing a greater understanding of the dynamics of training neural networks for reinforcement learning (Ceron & Castro, 2021; Araújo et al., 2021; Ostrovski et al., 2021; Schaul et al., 2022), the relationship between particular hyper-parameter configurations and performance on a given environment remains hard to predict.

One generally held desire in training neural networks is to reduce the variance of gradient updates, so as to avoid unstable and unreliable learning. In the reinforcement learning literature there has been a growing trend to use multi-step (or $n$-step) learning (Schwarzer et al., 2020; Agarwal et al., 2022) for improved performance. Despite their demonstrated advantage, researchers have been limited to small values of $n$ to avoid performance collapse, in part due to the increased variance arising from larger $n$. The supervised learning literature suggests that an effective mechanism for mitigating variance is through the choice of batch size: Shallue et al. (2019) empirically demonstrate that larger batch sizes result in reduced variance and increased performance. In this paper, we report the counter-intuitive finding that *reducing* the batch size can help avoid performance collapse with larger $n$-step updates. This is effectively doubling down on increased variance for improved performance. We also show that reduced batch sizes also results in reduced overall computation time during training.

## 2. Experimental Analysis

Advances in deep reinforcement learning (DRL) often build on prior algorithms, network architectures, and hyper-parameter selections. Given the large number of options, new work typically re-tunes only those components necessary for the new methods being considered. Thus, we have accumulated a set of, mostly static, parameters upon which new ideas are tested (this may be a form of the "social dynamics of research" hypothesized by Schaul et al. (2022)). One of the static parameters for training single-agent value-based agents has been the choice of batch size.

Since the introduction of DQN by Mnih et al. (2015), single-agent training on the Arcade Learning Environment (ALE, Bellemare et al., 2013) has used a batch size of 32, where this value was carefully tuned by the authors for performance. Since then, this value has rarely been changed, save for distributed agent training (Kapturowski et al., 2018; Espeholt et al., 2018). If one takes the general advice from the supervised learning literature, we should be aiming to increase the batch size so as to reduce variance and improve performance (Shallue et al., 2019). We focus on the effect of changing the batch size, while keeping all else equal. Check Appendix A in Appendix, for experimental setup details.

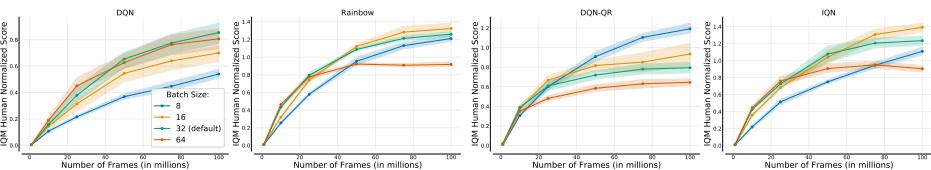

Figure 1: Varying batch sizes for DQN, Rainbow, QR-DQN, and IQN.

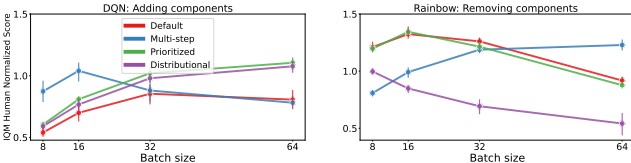

Figure 2: **Left:** Adding components to DQN; **Right:** Removing components from Rainbow.

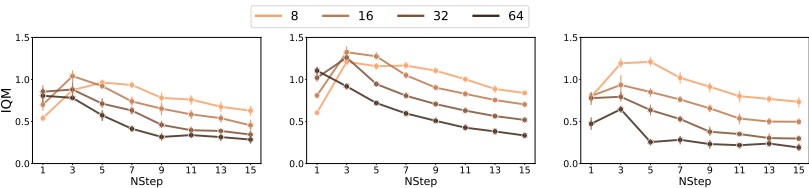

Figure 3: Varying batch sizes and $n$-steps in DQN (left), Rainbow (center), and QR-DQN (right).

**When do smaller batch sizes improve performance?**    We first varied the batch size for all agents (Figure 1). There are two surprising observations from this result. The first is that aggregate agent performance is relatively stable with respect to changing batch sizes. The second, and perhaps more surprising, is that *agent performance seems to improve with reduced batch size*. Indeed, we can observe that the default batch size is in fact not optimal for any of the agents and, with the exception of DQN, all agents seem to benefit from a *reduced* batch size.

The four agents considered differ in a number of respects. Three important considerations are that, of the 4, DQN is the only agent without distributional training (Bellemare et al., 2017), prioritized experience replay, and the only one without $n$-step returns. To get a better sense for whether either of these components is responsible for the reduced batch size performance boost, we performed ablation studies similar to those conducted by Ceron & Castro (2021). Since the version of Rainbow provided with the Dopamine library (Castro et al., 2018) is effectively DQN with three added components, we can investigate the changing dynamics as these components are added or removed from DQN and Rainbow, respectively. Figure 2 depicts the outcome of this ablation study. We find a striking pattern: while the four variants that use 1-step learning see their performance increase with greater batch sizes, as might be expected, the relationship is almost completely reversed for the variants using 3-step learning. Additionally, the other two components do not seem to present such a relationship with batch size.

The last results demonstrated there is a strong performance relationship between batch size and update horizon. We systematically explored this by evaluating various choices of these two parameters for three of the agents. As Figure 3 shows, the optimal batch size decreases as $n$ increases. This is most stark in QR-DQN, where simply reducing the batch size to 8 improves performance by close to 70% on the subset of games we consider.[1]  With Rainbow a batch size of 8 is able to maintain performance for $n$-step values as high as 9; in contrast, performance for the default batch size of 32 collapses beyond an $n$-step of 3.

## 3. Discussion

The long-term goal of RL research is to develop generally capable agents that can adapt to uncertain environments. Although theoretical results spanning multiple decades have given us a crisp insight into the mathematical properties of these algorithms, these theories unfortunately do not hold for non-linear function approximators such as neural networks. Given that neural networks have played a key role in the impact RL has had since 2015, it behooves the community to develop a better understanding of the the interplay of the various components and how changes can affect learning dynamics. Our work has revealed the striking finding that *doubling down* on variance by increasing $n$ and reducing batch size seems to, overwhelmingly so, produce improved performance. This flies in the face of traditional beliefs from the supervised learning community that reduced variance is best. Furthermore, it often results in substantial computational savings (see Appendix B).

---

[1]In Dopamine, QR-DQN uses an update horizon of 3 by default.

# 1 URM STATEMENT

The authors acknowledge that at least one key author of this work meets the URM criteria of ICLR 2023 Tiny Papers Track.

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

## A    EXPERIMENTAL SETUP:

For this case study, we use JAX implementations of agents provided by the Dopamine library (Castro et al., 2018) and applied to game-playing in the ALE (Bellemare et al., 2013).[2] For computational reasons, we evaluate our agents on 20 games chosen by Fedus et al. (2020) in their analysis of replay ratios; these were picked to offer a diversity of difficulty and dynamics. Similarly, we run each learning trial for 100 million frames (as opposed to the standard 200 million). The four agents we consider are: DQN (Mnih et al., 2015), Rainbow (Hessel et al., 2018)[3], QR-DQN (Dabney et al., 2018a), and IQN (Dabney et al., 2018b). These all use the default hyper-parameter values given in Dopamine. All experiments were run with 3 independent seeds on NVIDIA Tesla P100 GPUs. For evaluation, we follow the robust evaluation guidelines of Agarwal et al. (2021).

## B    COMPUTATIONAL CONSEQUENCES:

In deep reinforcement learning, improvements are typically evaluated based on sample efficiency, which refers to the number of interactions with the environment required to achieve a certain level of performance. However, this metric fails to account for differences in computational efficiency between algorithms. For example, two algorithms may have the same performance in terms of environment interactions, but one may take twice as long to complete each training step. In such cases, it would be preferable to choose the faster algorithm. Unfortunately, the DRL community often overlooks this important distinction in their standard evaluation methodologies.

Our results show that reducing batch size not only improves performance but also reduces computation time. Figure 4 illustrates that by using a smaller batch size, we can achieve better performance and do so in a fraction of the runtime. We encourage reader to revisit our findings above under this lens.

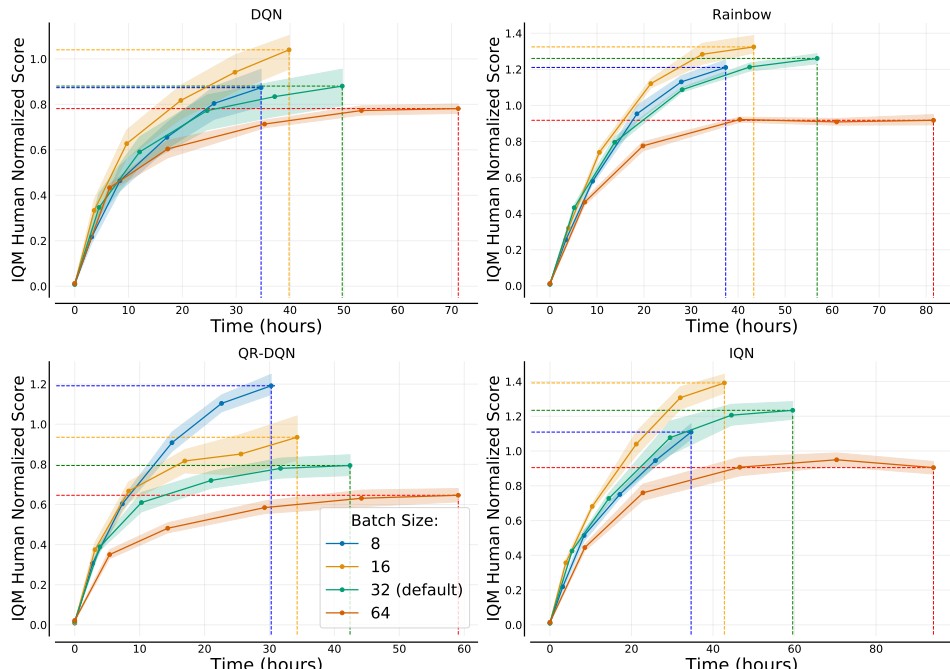

Figure 4: Measuring runtime versus performance when varying batch sizes in DQN, Rainbow, QR-DQN, and IQN (from left to right), all with $n$-step equal to 3.

---

[2]Dopamine uses sticky actions by default (Machado et al., 2018).
[3]Dopamine uses a "compact" version of the original Rainbow agent, including only $n$-step updates, prioritized replay, and categorical-distributional RL.

