# OpenReview forum: "The Small Batch Size Anomaly in Multistep Deep Reinforcement Learning"
_ICLR.cc/2023/TinyPapers — Submitted to Tiny Papers @ ICLR 2023_

### Official Review · Reviewer_Sz4v · 2023-03-19

**Confidence:** 3

**Summary Of Contributions:**

The paper studies the effect of the batch size on the family of DQN algorithms.

**Rating:**

Great Start (GS): a submission which meets some of the reviewing criteria but has room for improvement

**Strengths And Weaknesses:**


STRENGTHS
The study of the effects of batch size in the effectiveness of DRL is a big topic and can be beneficial for a wide community of researchers. The experiments are on the right direction and thorough ablation studies were performed on DQN.

WEAKNESSES

-  The main issue that I see with the paper and the reason for the low score is that there exist previous literature covering the same. For example a paper by Stooke and Abbeel: https://arxiv.org/pdf/1803.02811.pdf. It is unclear to the reader how the submitted paper fills a gap on the literature.

- Some claims of the paper are unsubstantiated, like when the paper says "the batch size of 32 is usually used". This is unreferenced and not entirely true.

- The conclusion of the paper states that: "our work has revealed the strike finding that doubling down on variance by increasing n and reducing the batch size seems to produce improved performance". However, it is not clear how the variance of the gradients update is related to the results shown in the experiments. The experiments show a different thing: the number of frames against the score for different batch sizes.

- The paper would benefit from distributing the code to make the experiments reproducible.








**Suggested Changes:**

1. It is recommended to try to include a short Abstract and it is customary to have a conclusion section at the end where you remind the reader of your main points and how you've proven them. Additionally, the conclusion section usually mentions some ways forward where the work can be extended. Maybe you can consider renaming the Discussion section to "Conclusions and Future work". This recommendation is aimed to align your work to the most common academic format. Also, your Discussion section is too verbose and Shakesperian, like out of style with academic writing. I would suggest re-writing it in a more academic style to economize space and have a few sentences for "future work". For example, sentences like: "the striking finding" or "it behooves the community to develop" can be shortened to conform into a more academic style of writing. All this sentence can be considered candidate for removal: "Although theoretical results spanning multiple decades have given us a crisp insight into the mathematical properties of these algorithms, these theories unfortunately do not hold for non-linear function approximators such as neural networks."

2. You mention twice that "The supervised learning literature suggests that an effective mechanism for mitigating variance is through the choice of batch size: Shallue et al. (2019) empirically demonstrate that larger batch sizes result in reduced variance and increased performance." However, I struggle to find that conclusion from the paper. In fact, a quick word search tells us that the word "variance" doesn't even appear on the cited paper. Consider finding another reference were the relationship between the gradient's variance and the batch size is more evident.

This paper by Stooke and Abeel can be a good reference: https://arxiv.org/pdf/1803.02811.pdf
In there, the claim is: "We further find it possible to train using batch sizes considerably larger than are standard, without negatively affecting sample complexity or final performance". In this paper, the batch size used for DQN is 512 (see figure 3). So your phrase "has used a batch size of 32, where this value was carefully tuned by the authors for performance. Since then, this value has rarely been changed, ..." seems not well supported.

Usually, the batch size is part of the hyper-parameter optimization.

3. There is a relationship between the batch-size and the learning rate ( https://arxiv.org/abs/1706.02677). Usually modern optimizers take care of this, but it is worth mentioning how you accounted for the learning rate change when the batch size is changed.

---

### Official Review · Reviewer_5vtn · 2023-03-30

**Confidence:** 4

**Summary Of Contributions:**

The authors study the intertwined impact of batch size and n-step learning on performance in deep RL. While reduced batch size and multi-step learning both increase the variance of learning, they find that combining the two helps increase the performance of the learned agent.

**Rating:**

High Potential (HP): a submission which meets the reviewing criteria and has potential to make an impact on the field

**Strengths And Weaknesses:**

## Strengths
- The authors have motivated the paper well and situated it in the context of prior work. The experimental setup and figures are easy to understand and follow. I agree that more research should be done into studying the effect of the different components of Rainbow instead of just accepting the current defaults as good enough.
- The experiments clearly back up the claims made by the authors.
    - Fig 2 isolates the components of Rainbow and shows that it is the addition of n-step learning that reverses the performance trend with respect to batch size in both Rainbow and DQN. Without n-step learning, both Rainbow and DQN show improved performance with larger batch sizes. And with n-step learning performance degrades with increased batch size.
    - In Fig. 3, across the methods shown, the smallest batch size is much more robust to increased values of n-steps and maintains performance while IQM scores degrades for larger batch sizes (32 & 64).
- There is enough detail in the paper to reproduce the main results. It would be better if the authors had also released the code for their experiments, but it is not a major concern since their experiments are based on Dopamine.

## Weaknesses
- The authors have not followed the formatting instructions for the Tiny Papers track (the style files for the text and headers are different from what the official template uses). They have also forgotten to include the abstract in the paper.
- I take issue with the strength of the statement made with respect to Fig 1 about “aggregate agent performance is relatively stable with respect to changing batch sizes. “ It appears to me that Figure 1 _does_ indeed show that performance varies quite a bit with different batch sizes. For each of the given methods, performance changes with different batch sizes, for example in DQN-QR, performance nearly doubles from a bs=64 to bs=8. I would suggest the authors revise this claim in the paper.


**Suggested Changes:**

- I would like to see the effect of varying batch size and n-steps on IQN too.
- More seeds would strengthen the paper, since in general papers in RL use 5 seeds.
- A future work that I would be interested in would be studying the gradient norms during training for varying batch sizes and n-steps and seeing if the doubling down of variance shows up in the gradient norms.

---

### Meta-Review · Area_Chair_qHLi · 2023-04-08

**Recommendation:** Invite to revise
**Confidence:** 4

**Metareview:**

The paper investigates the effect of batch size on the family of DQN algorithms and conducts thorough ablation studies. However, there exists previous literature on the same topic, and it is unclear how the paper fills a gap in the literature. Some claims in the paper are unsubstantiated, and the conclusion does not make clear the relationship between the variance of the gradients update and the experimental results. The paper would benefit from including an abstract, a conclusion section, and future work. Additionally, the discussion section should be rewritten in a more academic style to economize space, and some sentences should be considered for removal. The paper could also benefit from finding another reference that provides a more evident relationship between the gradient's variance and batch size. The cited paper by Shallue et al. does not support the claim made in the paper. Another reference provided by the reviewer, Stooke and Abeel's paper, could be a good reference. The paper could also consider the relationship between batch size and learning rate and how this was accounted for in the experiments. Overall, the paper is a great start but has room for improvement.

**Summary:**

The paper studies the effect of the batch size on the family of DQN algorithms.

**Comments And Feedback To The Authors:**

Please try to solve reviewers' concerns in the revised paper.

**Reason For Not Giving A Higher Recommendation:**

This paper is a good start but it remains a lot room for improvement. Firstly, this paper should follow the format of ICLR paper. An abstract is necessary or it is unfair for other paper authors since lack of abstract actually provides more space in considering the page limits. Second, the discussion section should be rewritten in a more academic style to economize space, and some sentences should be considered for removal. While the experimental setup is thorough and provides clear results, the paper does not sufficiently distinguish itself from existing literature on the topic.

**Reason For Not Giving A Lower Recommendation:**

N/A

---

### Decision · Program_Chairs · 2023-04-08

No revision received; not invited to archive